Non-SARS-CoV-2 genome sequences identified in clinical samples from COVID-19 infected patients: Evidence for co-infections

http://orcid.org/0000-0003-0109-6000 Abouelkhair Mohamed A. mabouelk@vols.utk.edu
Department of Biomedical and Diagnostic Sciences College of Veterinary Medicine, University of Tennessee , Knoxville, TN , USA
Krajaejun Theerapong
Electronic publication date: 2020 Nov 2
Publication date: 2020
Volume: 8
Electronic Location ID: e10246
Received 2020 Jun 19; Accepted 2020 Oct 6
Copyright: © 2020 Abouelkhair
Copyright year: 2020
Copyright holder: Abouelkhair
License: This is an open access article distributed under the terms of the Creative Commons Attribution License, which permits unrestricted use, distribution, reproduction and adaptation in any medium and for any purpose provided that it is properly attributed. For attribution, the original author(s), title, publication source (PeerJ) and either DOI or URL of the article must be cited.
License URL: https://creativecommons.org/licenses/by/4.0/

Keywords: COVID-19, Influenza A virus, Human immunodeficiency virus, Co-infections, Bacteria

Funding: The author received no funding for this work.

==============================
Background

In December 2019, an ongoing outbreak of pneumonia caused by severe acute respiratory syndrome coronavirus 2 (SARS-CoV-2/ 2019-nCoV) infection was initially reported in Wuhan, Hubei Province, China. Early in 2020, the World Health Organization (WHO) announced a new name for the 2019-nCoV-caused disease: coronavirus disease 2019 (COVID-19) and declared COVID-19 to be a Public Health Emergency of International Concern (PHEIC). Cellular co-infection is a critical determinant of viral fitness and infection outcomes and plays a crucial role in shaping the host immune response to infections.

Methods

In this study, 68 public next-generation sequencing data from SARS-CoV-2 infected patients were retrieved from the NCBI Sequence Read Archive database using SRA-Toolkit. Data screening was performed using an alignment-free method based on k-mer mapping and extension, fastv. Taxonomic classification was performed using Kraken 2 on all reads containing one or more virus sequences other than SARS-CoV-2.

Results

SARS-CoV-2 was identified in all except three patients. Influenza type A (H7N9) virus, human immunodeficiency virus, rhabdovirus, human metapneumovirus, Human adenovirus, Human herpesvirus 1, coronavirus NL63, parvovirus, simian virus 40, and hepatitis virus genomes sequences were detected in SARS-CoV-2 infected patients. Besides, a very diverse group of bacterial populations were observed in the samples.

Introduction

In December 2019, the first cases of coronavirus disease 2019 (COVID-19) were possibly due to a zoonotic transmission in China. It was tied to a large seafood market that also traded in live wild animals (Tay et al., 2020). The causative virus, severe acute respiratory syndrome coronavirus 2 (SARS-CoV-2), is capable of human-to-human transmission and rapidly spread to other regions of China and other countries (World Health Organization, 2020). It is now a global pandemic and is a considerable concern for public health. So far, more than 16,101,367 confirmed cases were diagnosed in nearly 213 countries and territories around the world and two international conveyances, causing globally over 645,000 deaths (Worldometer, 2020, July 25).

Coronaviruses are known to cause severe diseases in humans and animals. Of these, four human coronaviruses (229E, NL63, OC43, and HKU1) typically only infect the upper respiratory tract and cause relatively minor symptoms (Fehr & Perlman, 2015). However, there are three coronaviruses (severe acute respiratory syndrome coronavirus (SARS-CoV), Middle East respiratory syndrome coronavirus (MERS-CoV), and SARS-CoV-2) that can replicate in the lower respiratory tract and cause pneumonia, which can be fatal. With 79% genome sequence similarity, SARS-CoV is the closest relative to SARS-CoV-2 among human coronaviruses (Gorbalenya et al., 2020). However, of all known coronavirus sequences, SARS-CoV-2 is most similar to bat coronavirus RaTG13, with a similarity of 98% (Zhou et al., 2020).

SARS-CoV-2 pathophysiology closely parallels that of SARS-CoV infection, with active inflammatory responses strongly implicated in the resulting airway damage (Wong et al., 2004). Hence the extent of the disease in patients is attributed not only to the viral infection but also to the host’s response (Tay et al., 2020).

Underlying co-infections in primary infectious disease are an important variable that needs to be considered but is often undetected. Remarkable developments in next-generation sequencing have recently made metagenomics, an unbiased shotgun method of analysis, a widely used tool in just about every field of biology, including diagnosis of infectious diseases (Kuroda et al., 2012; Lecuit & Eloit, 2014). Metagenomics is powerful because it can diagnose unsuspected microbial agents (Wilson et al., 2014). It directly analyzes samples in their entirety, eliminating the need for prior knowledge to obtain comprehensive information. In this capacity, metagenomics exceeds traditional diagnostic limitations. With the microbial genomes in hand, we can now explore the possibility of using metagenomic and metatranscriptomic next-generation sequencing (mNGS) directly as a screening method for detection of other microbes in clinical samples (Plyusnin et al., 2020).

A simple approach would be to first map sequencing reads from the sample to the reference microbial genome. However, the accuracy of such an alignment-based method is relatively low compared to an alignment-free approach. In the alignment-based method, genome sequences from closely related viruses can lead to false-positive results (Chen et al., 2020b). Moreover, the virus-specific reads obtained may not be adequate for unambiguous detection (degraded RNA), incompletely target-enriched sequence library by multiple-PCR (Lundberg et al., 2013) or hybrid capture (Duncavage et al., 2011), which can lead to false-negative results.

Fastv is an alignment-free, ultra-fast tool for detecting the microbial sequences in sequence data (Chen et al., 2020b). It can identify target microorganisms using unique k-mers. It detects SARS and other coronaviruses from sequencing data and efficiently distinguishing SARS from MERS.

In this study, public next-generation sequencing data from SARS-CoV-2 infected patients were analyzed by fastv using the pre-computed unique k-mer resources (Chen et al., 2020b). Taxonomic classification was performed using Kraken 2 on all reads containing more than one microbial sequence (Wood, Lu & Langmead, 2019). The present study’s findings have confirmed the actual existence of genome sequences of other microbes in SARS-CoV-2 infected patients.

Materials and Methods

Computing hardware

Amazon Elastic Compute Cloud (EC2) instance (i.e., virtual server in the AWS cloud) was used. SRA-tools package version 2.9.1 (https://github.com/ncbi/sra-tools), Kraken 2 (https://github.com/DerrickWood/kraken2) (Wood, Lu & Langmead, 2019), and fastv version 0.9.0 (https://github.com/OpenGene/fastv) (Chen et al., 2020b) were installed within the Linux 2 EC2 instance.

SRA database mining

Next-generation sequencing technologies have enabled large-scale genomic surveillance of SARS-CoV-2 as thousands of isolates are being sequenced worldwide and deposited in public data repositories. The sequence data were downloaded as .sra files using the prefetch tool (https://github.com/ncbi/sra-tools/tree/master/tools/prefetch), then extracted to .FASTQ files using the NCBI fastq-dump tool (https://github.com/ncbi/sra-tools/tree/master/tools/fastq-dump). Data sets for analysis were chosen through a keyword search of the SRA descriptions for “COVID19” and downloaded between 27 January 2020 and 16 May 2020. Sequence data from negative COVID-19 patients, experimental studies, and controlled access were excluded (Table S1).

Sequence data pre-processing and screening using fastv

Fastv, along with the pre-computed unique k-mer resources, was used as previously described (Chen et al., 2020b). Initially, fastv performed adapter trimming, quality pruning, base correction, and other pre-processing to ensure the accuracy of k-mer analysis on fastq input files. The fastv tool identifies a target virus from sequencing data and detects any microbial sequences for which a unique k-mer data is provided. A unique SARS-CoV-2 k-mer set and the SARS-CoV-2 reference genome were used as input files (downloaded from https://github.com/OpenGene/fastv/tree/master/data) with the k-mer collections for viral and microbial genomes (downloaded from http://opengene.org/microbial.kc.fasta.gz). The k-mer scanning results were visualized in a figure on a single HTML page by fastv.

Sequence data analysis using Kraken 2

The results of fastv were validated with Kraken 2. Kraken 2 is the latest version of Kraken, a taxonomic classification system that uses exact k-mer matches to achieve high accuracy and rapid classification (https://github.com/DerrickWood/kraken2) (Wood, Lu & Langmead, 2019). Kraken constructs an index of all k-mers found in the reference genomes and assigns each k-mer to the least common ancestor (LCA) of all species that have that k-mer. Then Kraken matches the k-mers contained in the reads to this index and eventually assigns the reads to the taxon with the most fitting k-mers by following the path from the root of the tree (Wood, Lu & Langmead, 2019). Through lowering memory use by 85%, Kraken 2 improves upon Kraken 1, enabling higher numbers of genomic reference data to be used while retaining high accuracy and fivefold speed. A standard database containing RefSeq complete bacterial, archaeal, and viral genomes, along with the human genome and a collection of known vectors (UniVec_Core), was downloaded (https://ccb.jhu.edu/software/kraken2/downloads.shtml). The database was then constructed using 32 threads with the default parameters on an AWS EC2 h1.8xlarge storage optimized instance with 16 dual-core hyperthreaded 2.30 GHz CPUs and 132 gigabytes (GB) of RAM.

Data visualization

The results of Kraken 2 analysis were visualized with Pavian (https://github.com/fbreitwieser/pavian) (Breitwieser & Salzberg, 2016) and Krona tool (https://github.com/marbl/Krona/wiki) (Ondov, Bergman & Phillippy, 2011), which displays hierarchical data (like taxonomic assignation) in multi-layered pie charts. The Kraken 2 outputs were converted in HTML format using the program ktImportTaxonomy (https://github.com/marbl/Krona/tree/master/KronaTools/scripts), which parses the information relative to the query ID and the taxonomy ID.

Results

SARS-CoV-2 identification

Sequence data analysis was performed on public Illumina HiSeq/MiSeq libraries from the NCBI SRA database (Bioproject PRJNA605983) sequenced from bronchoalveolar lavage fluid from five patients (WIV02, WIV04, WIV05, WIV06, and WIV07) with pneumonia at the early COVID-19 outbreak in Wuhan, China. Nine libraries were downloaded as .sra files using the prefetch tool; then, the fastq files were extracted using the NCBI fastq-dump tool. Prior to the k-mer analysis, sequencing adapters and low-quality bases were removed by fastv. After scanning the fastq data, fastv reported the k-mer coverage for each microbial genome with valid hits (Fig. 1).

Figure 1 SARS-CoV-2 detection by fastv.

SARS-CoV-2 was detected in WIV07 patient, where twelve SARS-CoV-2 hits were included in the genome list and ordered by k-mer coverage (99.4972–99.2308%). The number of hits (on the y-axis) of each k-mer key (on the x-axis) were plotted. Mismatches were highlighted in red.

SARS-CoV-2 was detected in all of the 68-sequence data except SRR11772662, SRR11772663, and SRR11772664 samples were negative for SARS-CoV-2. These samples belong to one study (Bioproject PRJNA631042), where the research group used different sequencing technologies on the same sample to find a cost-effective and highly scalable method for SARS-CoV-2 sequencing. Because sequence technologies vary in reading depth and coverage thresholds, fastv could not detect SARS-CoV-2 in sequenced samples with lower coverage metrics.

Non-SARS-CoV-2 genome sequences were detected in COVID-19 infected patients

Fastv also identified influenza type A (A/Shanghai/02/2013(H7N9)) and rhabdovirus genomic sequences in the data from WIV02 (SRR11092058 and SRR11092063 data), WIV04 (SRR11092057 and SRR11092062 data), WIV05 (SRR11092061 data), WIV06 (SRR11092056 and SRR11092060 data), and WIV07 (SRR11092059 and SRR11092064 data) patients.

Influenza type A segment 4 hemagglutinin (HA) gene was detected with high coverage (100%, 100%, 98.57%, 95.71%, and 100%) and mean depth (5.58, 16.52, 4.8, 2.95, and 881.37) in WIV02, WIV04, WIV05, WIV06, and WIV07 patients, respectively. Genes coding for Influenza A virus polymerase, non-structural proteins, matrix proteins 1 and 2 were also identified in the previously mentioned patients but with lower coverages. The genome sequence of rhabdovirus was detected with coverage (57.36%, 72.73%, 57.58%, 67.75%, and 53.46%) and mean depth (4.82, 8.13, 4.84, 5.08, and 4.06) in WIV02, WIV04, WIV05, WIV06, and WIV07 patients, respectively.

The genome sequence of the Nipah virus was detected in WIV05, WIV06, and WIV07 patients. Infection with the Nipah virus in humans causes a number of clinical manifestations that range from asymptomatic (subclinical) infection to acute respiratory infection and fatal encephalitis.

In addition, human immunodeficiency virus, human herpesvirus 1, human T-lymphotropic virus 1, hepatitis virus, and simian virus 40 were found in WIV07 patient sequence data with coverage below 21%. The low coverage and non-human specific pathogen hits have been overlooked (Table S2).

Human coronavirus 229E genome sequence was detected (coverage = 14% and mean depth = 1.55) in SRR11772654 (Bioproject PRJNA631042). Human adenovirus 5 sequences were detected in SRR11772660, SRR11772663, SRR11772666, SRR11772672, SRR11772675 and SRR11772680 data. In addition, human adenovirus 1 sequences were detected in SRR11772663 and SRR11772666 data. Parvovirus NIH-CQV genes coding for a putative replication-associated protein (rep), and putative capsid protein (cap) (coverage = 14.51% and mean depth = 0.41) were detected in SRR10971381 (Bioproject PRJNA603194) (Table S2).

A very diverse group of bacterial populations were observed in the COVID-19 infected patients. Enterobacter hormaechei was identified in WIV02, WIV04, WIV05, WIV06, and WIV07 patients. Enterobacter hormaechei is an important emerging pathogen and can cause nosocomial infections, and often have resistance to multiple clinically relevant antibiotics (Monahan et al., 2019). Acinetobacter baumannii sequence was detected in WIV05 and WIV07 patients. A. baumannii is one of the most successful pathogens associated with hospital-acquired infections worldwide (Lee et al., 2017). Enterococcus faecalis sequence was identified in WIV04, WIV05, and WIV07 patients (Table S2).

Coliphage phi-X174 was identified in most sequence data, which might be introduced by the Illumina PhiX control library (Meyer & Kircher, 2010). Stenotrophomonas phage phiSMA7, Enterobacteria phage phi80, DE3, Fels-2 and M13, Proteus virus, and Delftia phage RG-2014 were identified in the sequence data (Table S2).

Kraken taxonomic classification

Taxonomic classification was performed using Kraken 2 on all reads containing more than one virus sequences. The genetic data for constructing the databases were retrieved from the NCBI RefSeq library. A very diverse group of viral, bacterial, and archaeal populations was observed in the samples. A taxonomic classification that was obtained from WIV04, WIV06, and WIV07 patients revealed a dominance of Bacteria (7%, 5%, and 59%, respectively), followed by Viruses (0.3%, 0.3%, and 0.3%, respectively) while Archaea was lower than 0.02% in all patients. Among viral communities, influenza type A (8%), rather than SARS-CoV-2 (2%), was found to be dominant in the WIV07 patient, which is consistent with fastv result (Fig. 2). SARS-CoV-2, rhabdovirus, and influenza type A dominated the sequence data from WIV06 patient (Fig. 3) and WIV04 patient (Fig. 4).

Figure 2 The Krona pie chart of the viral taxonomy of the WIV07 patient.

Viruses identified from kraken 2 analysis of SRR11092059 data were visualized using krona tool. Among viral communities, influenza type A (8%), SARS-CoV-2 (2%) and rhabdovirus (0.6%) dominated the sequence data from WIV07 patient.

Figure 3 The Krona pie chart of the viral taxonomy of the WIV06 patient.

Viruses identified from kraken 2 analysis of SRR11092060 data were visualized using krona tool. Among viral communities, SARS-CoV-2 (1%), rhabdovirus (0.6%) and influenza type A (0.4%) dominated the sequence data from WIV06 patient.

Figure 4 The Krona pie chart of the viral taxonomy of the WIV04 patient.

Viruses identified from kraken 2 analysis of SRR11092062 data were visualized using krona tool. Among viral communities, SARS-CoV-2 (1%), rhabdovirus (0.8%) and influenza type A (0.4%) dominated the sequence data from WIV04 patient.

A rank code, indicating unclassified (U), domain (D), kingdom (K), phylum (P), class (C), order (O), family (F), genus (G), or species (S) was used.

Acinetobacter baumannii and Enterobacter species genomic sequences were detected in SRR11092059 (Fig. 5), SRR11092060 (Fig. 6), SRR11092061, SRR11092062 (Fig. 7), and SRR11092063 sequence data. By using Kraken2, a tool for read taxonomy, many bacterial species were identified that most of them present in the normal flora and rarely cause problems. However, compared to the fastv result, the Pasteurella multocida sequence was detected in SRR11092062 (Fig. 7) and SRR11092063 sequence data. The genome sequence of Staphylococcus aureus was detected in SRR11092059 data (Fig. 5).

Figure 5 Bacteria identified in SRR11092059.

Sankey diagrams of Kraken 2 report results obtained from SRR11092059. The width of the flow is proportional to the number of reads. The number above each node is the number of k-mer hits. A rank code, indicating domain (D), phylum (P), family (F), genus (G), or species (S) was used.

Figure 6 Bacteria identified in SRR11092060.

Sankey diagrams of Kraken 2 report results obtained from SRR11092060. The width of the flow is proportional to the number of reads. The number above each node is the number of k-mer hits. A rank code, indicating domain (D), phylum (P), family (F), genus (G), or species (S) was used.

Figure 7 Bacteria identified in SRR11092062.

Sankey diagrams of Kraken 2 report results obtained from SRR11092062. The width of the flow is proportional to the number of reads. The number above each node is the number of k-mer hits. A rank code, indicating domain (D), phylum (P), family (F), genus (G), or species (S) was used.

The most abundant bacterial phylum was Bacteroidetes, followed by Firmicutes phylum in SRR10971381 sequence data (Fig. 8).

Figure 8 Sankey diagrams of Kraken 2 report results obtained from SRR10971381.

The width of the flow is proportional to the number of reads. The number above each node is the number of k-mer hits. A rank code, indicating domain (D), kingdom (K), phylum (P), family (F), genus (G), or species (S) was used.

Discussion

The importance of detecting co-infections is becoming more recognized (Griffiths et al., 2011; Li & Zhou, 2013), but it remains challenging to get such information. The source of clinical samples and the sequencing technology can be inferior in co-infection detection (Birdsell et al., 2018).

A better understanding of co-infection’s prevalence is required, partly because co-infecting pathogens can interact with each other directly or indirectly via the host’s resources or immune system (Cox, 2001; Griffiths et al., 2011). These interactions within co-infected hosts can alter the transmission, clinical progression, and control of multiple infectious diseases as compared to single pathogen species infection (Chiodini, 2001; Griffiths et al., 2011; Palacios et al., 2009). Recent studies indicated that co-infection’s adverse effects are more common than those with no-effects or positive impacts on human health (Pullan & Brooker, 2008).

The underdiagnosis of co-infections is attributed to a lack of clinical suspicion, common symptoms, and the fact that conventional methods have little capacity to detect co-infections in the absence of a priori knowledge.

Co-infection can potentially affect the performance of laboratory testing for coronavirus disease 2019. A previous study (Lai, Wang & Hsueh, 2020) showed that reverse transcription polymerase chain reaction (RT-PCR) could not detect SARS-CoV-2 in two patients co-infected with influenza A virus. Some researchers have suggested that inadequate viral specimens, the use of improperly validated assay, timing and methods of collecting specimens, the presence of mutations at the primer binding site, and co-infection with other viruses might be responsible (Arevalo-Rodriguez et al., 2020; Kucirka et al., 2020; Lai, Wang & Hsueh, 2020; Li et al., 2020). Based on a limited number of observational studies, it was found that the false-negative rate for SARS-CoV-2 RT-PCR was 20% (Lai, Wang & Hsueh, 2020; Li et al., 2020).

Given these considerations, COVID-19 might be underdiagnosed, especially during the influenza season, since typical clinical symptoms of COVID-19, including fever, cough, and dyspnea, resemble those of influenza (Chen et al., 2020a; Wu et al., 2020).

Therefore, exploring new diagnostic approaches is essential to advance understanding of co-infection contribution to disease manifestations and treatment responses (Birdsell et al., 2018).

Remarkable developments in next-generation sequencing have recently made metagenomics, an unbiased shotgun method of analysis, a widely used tool in just about every field of biology, including diagnosis of infectious diseases (Kuroda et al., 2012; Lecuit & Eloit, 2014).

In this study, using k-mer based tools (fastv and Kraken 2), genome sequences of various microorganisms, including the SARS-CoV-2 virus, were detected. The accuracy of such an alignment-free method is relatively high compared to an alignment-based approach. Fastv is an alignment-free, ultra-fast tool for detecting the microbial sequences in sequence data using unique k-mers (Chen et al., 2020b). To validate the results obtained from fastv analysis, Kraken 2 was used. Kraken 2 is the latest version of Kraken, a taxonomic classification system that uses exact k-mer matches to achieve high accuracy and rapid classification (Wood, Lu & Langmead, 2019).

Kraken 2 is fast and requires less memory than Kraken 1. However, the standard database construction requires approximately 100 gigabytes (GB) of disk space and random-access memory (RAM) of more than 30 GB. For that purpose, an AWS EC2 h1.8xlarge storage optimized instance with 16 dual-core hyperthreaded 2.30 GHz CPUs and 132 GB of RAM was used.

Multidrug-resistant bacteria associated with hospital-acquired infections worldwide, such as E. hormaechei , S. aureus, P. multocida, and A. baumannii, were detected in COVID-19 infected patients (Lee et al., 2017; Monahan et al., 2019). Most of the infections caused by the previously mentioned bacteria occur in critically ill and or immunocompromised patients in the intensive care unit (ICU) setting (Fournier, Richet & Weinstein, 2006).

Understanding the nature and consequences of co-infection is essential for accurate estimates of infectious disease burden. In particular, more systematic data on infectious diseases would also help measure the extent of co-infection on human health. Increased knowledge of the risk factors, the conditions in which co-infecting pathogens interact, and the mechanisms behind these interactions, particularly in experimental studies, will also help develop and evaluate infectious disease management programs.

Conclusions

In this study, screening of 68 public next-generation sequencing data from SARS-CoV-2 infected patients was performed using fastv and Kraken 2. Multiple viruses, including the SARS-CoV-2 virus, genome sequences were detected. Further large-sample studies are warranted to investigate the prevalence of COVID-19 co-infection, the impact of co-infection on the host immune system of COVID-19 patients, and their role in disease progression.

Supplemental Information

Supplemental Information 1 SRA sequences used in this study with the detection result for SARS-CoV-2 k-mer.

Click here for additional data file.

Supplemental Information 2 Identified viral and bacterial genomes sequences by fastv.

Click here for additional data file.

Additional Information and Declarations

Competing Interests

Author Contributions

Data Availability

The author declares that they have no competing interests.

Mohamed A Abouelkhair conceived and designed the experiments, performed the experiments, analyzed the data, prepared figures and/or tables, authored or reviewed drafts of the paper, and approved the final draft.

The following information was supplied regarding data availability:

The sequences are available at Genbank: SRR11772680, SRR11772675, SRR11772672, SRR11772666, SRR11772663, SRR11772660, SRR11772654, SRR11772640, SRR11537952, SRR11092064, SRR11092063, SRR11092062, SRR11092061, SRR11092060, SRR11092059, SRR11092058, SRR11092057, SRR10971381, SRR11092056.

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
