# Peer review of "Non-SARS-CoV-2 genome sequences identified in clinical samples from COVID-19 infected patients: Evidence for co-infections"

_PeerJ, doi:10.7717/peerj.10246_

## Round 0.1 · original submission · Major Revisions

Two reviewers, who are experts in the field, have evaluated your manuscript, and their comments are attached here. The reviewers raise critical concerns that the current manuscript lacks novelty, essential methodology (i.e., inclusion criteria for selecting the SRA data, data analysis by Fastv program), result presentation and validation (including Tables and Figures), thorough discussion (i.e., significance and relevance of co-infection), and appropriate references. The manuscript also needs to improve readability (i.e., typos, suitable terms, and grammar). The Fastv outputs should be publicly available, so other investigators could access and re-use the data. The author should emphasize on how the results of this study add up to the existing knowledge. If you decide to revise the manuscript, please respond to the reviewers' comments point-by-point. A substantially-revised manuscript is required for further consideration for publication.

Reviewer 1 ·

Basic reporting

In this study, the author examined infection of COVID-19 and other viruses in publicly available 68 SRA data using Fastv program, which is published in bioRxiv:2020.2005.2012.092163. 10.1101/2020.05.12.092163. Author identified infection of COVID-19 in 65 SRA data. In addition, several viruses were identified.

Experimental design

In fact, this manuscript should be rejected due to several reasons. The first is that there is no novelty of this study. Similar study is already done by Chen et al. who developed Fastv program. The difference is just the number of SRA data for virus identification. Although the author increased the number of SRA data for virus identification, I cannot find any valuable results. Title should be also changed. Most SRA data were derived from transcriptome data not metagenome data. Maybe co-infection might be common in COVID-19 infected samples. I cannot find any results which support the identification of other viruses in the examined SRA data.

Validity of the findings

There is no result associated with validation of the findings in this study.

Additional comments

The manuscript is really poorly written. There are lots of errors in English. Many sentences and paragraphs were poorly written. The materials and methods were not sufficiently written. I only see the detailed information of Macbook pro which was used for data analysis. The results were really badly written. Without proper results, how the author can write the discussion part?

There were lots of errors. These are some examples.
In abstract
L32: sixty-eight public next-generation sequencing libraries -> sixty-eight public next-generation sequencing data
L34: Screen the raw reads for viral genome sequences, including SARS-COV-2, using an alignment-free method based on K-mer mapping and extension. -> The sentence is not complete.

L65 (Viruses 2020) -> The reference is not correct.
L66 98 percent -> 98%
L104 A reference for Fastv is required.
L109 libraries -> data

Materials and methods
L114-118 The hardware for data analysis may be not necessary.
L120-126 Authors selected SRA data from six projects. Which thresholds were used to select SRA data?
L126 Table. 1 -> Table 1
L 128-138 It is necessary how authors conducted data analysis using Fastv program in detail.

L142 We conducted -> I conducted (there is only one author in this manuscript)
The Table 1 is not informative. Author should describe sample information for individual SRA data.
L149-150 No information associated with PRJNA631042 is available.
Results and discussion are terribly written. Author should write results in detail.
Tables and figures were poorly prepared.

Reviewer 2 ·

Basic reporting

This paper reported identification of possible viral co-infections in samples from SARS-CoV-2 infected patients. This article is well written and easy to follow. However, Figures and Tables need a major improvement. In general, the results section was quite vague where random outputs from Fastv software are shown.

Experimental design

The author simply used Fastv to analyse 68 publicly available SARS-CoV-2 samples from SRA database. Fastv outputs for all analysed samples are not provided. The author should make the Fastv output for all samples publicly available.

Validity of the findings

Whilst the authors discovered possible co-infections in a number of samples, the significance of these identification as well as their relevance for the accurate estimates of infectious disease burden was not explored. The authors should either attempt to describe how these co-infections could be driving disease progression or outcome, or outline that the consequence of these co-infections are still unclear.

Additional comments

Table 1 should be placed in supplementary. Can also add more information about each sample such as read length, total sequences, total bases, etc.

Figure 1's resolution is too low. Impossible to read.

Remove "Figure.a:" from Figure 1 text.

Which sample does Figure 1 represent?.

Line 158: Figure.2, a: should be Figure 1

Line 161: Figure.2, b: there is no description of this Figure in the manuscript.

Line 166-168: The author should provide sampleIDs for those possible co-infected samples.

Line 168: Figure 3. Why use Krona to visualise these 3 samples? Are they the only samples that contain possible co-infection?

For all co-infected samples, what are their coverage and depth? Can they be false positive?

---

## Round 0.2 · Major Revisions

The revised manuscript is markedly improved. However, both reviewers comment on that some major issues raised in the first round of review are not yet addressed. Besides, the reviewers provide a list of additional concerns to improve the manuscript (i.e., data presentation and analysis, readability of the manuscript, adding figure(s), some data/tables/figures/links are missing). So, the manuscript has now been returned for major revision. If you decide to go for revision, please address the reviewers' concerns point-by-point, before the manuscript undergoes further consideration for publication in PeerJ.

Reviewer 1 ·

Basic reporting

In this study, the author examined infection of COVID-19 and other viruses in publicly available 68 SRA data using Fastv program and Kranken 2.

Experimental design

As compared to the previous version of the manuscript, the revised manuscript is very much improved. However, author did not provide many results derived from the data analysis such as bacterial taxonomy, bacteria phages, and non-SARS-CoV-2.

Validity of the findings

Author used two different programs for the virus identification. Comparative analysis might be useful.

Additional comments

Authors used two different programs for the virus identification. Comparative analysis might be useful.

L95 identification experiments -> Not clear to understand. Please revise it.
L96 using fastv, along with -> by fastv using the pre-computed ~

Supplementary data
Fastv results such as peerj-50170-SRR10971381.html file should be summarized. Please do not show the raw results as it is.

L133 We validated -> I validated (There is only one author in this manuscript. Check it thoroughly in the manuscript.)
L183 Author stated that coverages ranged from 47.55% to 53.22% indicating that the obtained sequences for the rhabdovirus is not complete genome. Please revise it.
L208 Please revise the sentence as follows.
A rank code, indicating unclassified (U), domain (D), kingdom (K),
In the results, I would like to suggest generating phylogenetic trees using obtained viral sequences.

L228 Please revise the sentence. It is not easy to understand.
The value of identifying underlying co-infection(s) is gaining greater appreciation
L232 urgently required -> delete urgently
L253-260 It might be deleted.
In discussion, it is necessary to discuss about identified bacteria.
Authors used two different programs for the virus identification. Please compared the difference between two different methods in the results and discussion.

Figure 2-4 Author identified many reads associated with bacteria phages. Please write results and discussion about the identified bacterial phage.
I can not find any tables or figures addressing the identified non-SARS-CoV-2. Please include them.
In addition, author should provide a table or a figure showing identified bacteria by Kranken2.

Reviewer 2 ·

Basic reporting

The author has addressed some of the concerns I previously raised. However, more information is still needed at various places in the manuscript.

Experimental design

Download links and references for all the tools and databases used in this study need be added to the manuscript. All parameters also to be shown. This is to ensure reproducibility.

Validity of the findings

All analysis outputs have to be provided to show the validity of the findings described in this manuscript

Additional comments

LINE 241: "SRA-tools package (version 2.9.1), Kraken 2, and fastv (version 0.9.0)"
- Please add download links and references for these tools.

LINE 247-248: "The sequence data were downloaded as .sra files using the prefetch tool, then extracted to. fastq files using the NCBI fastq-dump tool. "
- Please add download links and references for these tools.

LINE: 285 "a k-mer-based program "
- Please elaborate. A k-mer-based program for?

LINE 292-294 "A standard database containing RefSeq complete bacterial, archaeal, and viral genomes, along with the human genome and a collection of known vectors (UniVec_Core) was downloaded."
- Please add download links for these datasets.

LINE 294: "Then the database was built using"
- Please provide database building parameters (e.g. default parameters?)

LINE 311-312: "We converted the Kraken 2 output in HTML format using the program ktImportTaxonomy"
- Please add the download link for this tool.

LINE 326-328: "Figure 1: SARS-CoV-2 detection by fastv. SARS-CoV-2 was detected in WIV07 patient, where twelve SARS-CoV-2 hits were included in the genome list and ordered by k-mer coverage (99.4972% to 99.2308%). Mismatches were highlighted in red. "
- Provide more description. What are x and y axes representing?

LINE 330: "SARS-CoV-2 was detected in all of the other 68-sequence date"
- date -> data
- all of the other 59 sequence data

LINE 330-331: "SARS-CoV-2 was detected in all of the other 68-sequence date except three samples (Bioproject PRJNA631042) were negative for SARS-CoV-2. "
- Which sample/run ID were negative?

LINE 430-432: "Influenza type A (8%), rather than, SARS-CoV-2 (2%) was found to be dominant in the WIV07 patient (Figure 2)."
- Is this consistent with Fastv result?

SECTION "Kraken Taxonomic classification"
- Provide results for all samples

LINE 472-474 "A previous study (Lai et al. 2020) showed that reverse transcription polymerase chain reaction (rRT -PCR) could not detect SARS-CoV-2 in two patients co-infected with influenza A virus."
- Why cannot detect?
- How often does this happen?


Sample SRR11772648 that is provided as supplementary data is not found in Table S1.

Only 5 Fastv outputs are provided as supplementary data. The author should make all 68 Fastv outputs available.

From the 5 Fastv outputs that are provided as supplementary data, each output appears to be generated using slightly different Fastv parameters. The author should be consistent with how Fastv is applied to all of the 68 datasets. This is to ensure that all of the results are comparable and without biases.

Here are the commands that were used to generate the 5 outputs provided.

./fastv -i SRR10971381.fastq -c viral.kc.fasta.gz
./fastv -i SRR11092059.fastq -c viral.kc.fasta.gz -g SARS-CoV-2.genomes.fa -k SARS-CoV-2.kmer.fa
./fastv -i SRR11092062.fastq -c Downloads/microbial.kc.fasta.gz
./fastv -i SRR11092062.fastq -c viral.kc.fasta.gz
./fastv -i ncbi/public/sra/SRR11772648.fastq -c Downloads/microbial.kc.fasta.gz -w 10

---

## Round 0.3 · accepted · Accept

The author has properly addressed the concerns of both reviewers. The manuscript is now suitable for publication in PeerJ.

Please note that the author is required to correct a few typos in Lines 129 and 381 of the revised manuscript.

Reviewer 1 ·

Basic reporting

The author properly revised the manuscript according to reviewers' comments.

Experimental design

The author properly revised the manuscript according to reviewers' comments.

Validity of the findings

The author properly revised the manuscript according to reviewers' comments.

Additional comments

The author revised the manuscript according to reviewers' comments.
I suggest the manuscript for publication.

Reviewer 2 ·

Basic reporting

I would like to thank the authors for carefully revising the manuscript. I believe all my concerns have been properly addressed.

Experimental design

All my concerns regarding Experimental design have been addressed.

Validity of the findings

All my concerns regarding findings of this study have been addressed.

Additional comments

Here are a few more errors I came across in the manuscript

Line 129 "extracted to. fastq files"
- Remove . next to 'to'

Line 381 "detected in SRR11092062 (Figure 7) and SRR11092063 sequence date."
- Change 'date' to 'data'.